# Differential Impacts of Cereal and Protein Sources Fed to Pigs after Weaning on Diarrhoea and Faecal Shedding of *Escherichia coli*, Production, and Total Tract Apparent Digestibility

**DOI:** 10.3390/ani13050863

**Published:** 2023-02-27

**Authors:** John R. Pluske, Bruce P. Mullan, Jae Cheol Kim, David J. Hampson

**Affiliations:** 1College of Environmental and Life Sciences, Murdoch University, Murdoch, WA 6150, Australia; 2School of Agriculture and Food, Faculty of Science, The University of Melbourne, Parkville, VIC 3010, Australia; 3Department of Primary Industries and Regional Development, Perth, WA 6000, Australia; 4CJ BIO APAC, 622 Emporium Tower, Bangkok 10110, Thailand

**Keywords:** pigs, weaning, rice, performance, diarrhoea, digestibility

## Abstract

**Simple Summary:**

Newly weaned pigs are typically fed combinations of cereals and proteins to maximise performance. In the absence of the use of certain antimicrobial compounds, combinations of cereals and protein sources can also be used strategically to reduce dysbiosis in the gastrointestinal tract. This experiment examined the impacts of offering either medium-grain or long-grain extruded rice or wheat, in combination with animal or vegetable protein sources, on postweaning performance, shedding of β–haemolytic *Escherichia coli*, and the coefficient of total tract apparent digestibility (CTTAD) in the 21 days after weaning. The experimental findings confirmed that extruded rice is an excellent cereal for young pigs but that vegetable protein sources decreased production in weeks two and three compared to the use of animal protein sources. Vegetable protein sources decreased the faecal *E. coli* score. The CTTAD of dietary components differed according to interactions between cereal and protein sources.

**Abstract:**

Different cereal types, in combination with different protein sources, are fed to pigs after weaning, but their interactions and possible implications are not well researched. In this study, 84 male weaned piglets were used in a 21-day feeding trial to investigate the effects of feeding either medium-grain or long-grain extruded rice or wheat, in a factorial combination with protein sources of either vegetable or animal origin, on postweaning performance, shedding of β–haemolytic *Escherichia coli*, and the coefficient of total tract apparent digestibility (CTTAD). Pigs fed either rice type performed the same (*p* > 0.05) as wheat-fed pigs after weaning. The use of vegetable protein sources reduced growth rate (*p* < 0.001) and feed intake (*p* = 0.007) and deteriorated the feed conversion ratio (*p* = 0.028) in weeks two and three compared to pigs fed animal protein sources. The number of antibiotic treatments given for clinical diarrhoea was similar *(p* > 0.05). However, the faecal *E. coli* score showed a trend for the main effect of protein source, with pigs fed animal proteins showing a higher *E. coli* score than pigs fed vegetable proteins (0.63 vs. 0.43, *p* = 0.057). There was also a tendency for an interaction (*p* = 0.069) between cereal type and protein source (*p* = 0.069), with this difference being associated with a greater faecal score in pigs fed diets with long-grain rice plus animal proteins and wheat plus animal proteins. Significant interactions occurred for the CTTAD when assessed in week three. In general, pigs fed diets with medium-grain rice or long-grain rice with animal proteins had a higher (*p* < 0.001) CTTAD for dietary components than pigs fed all other diets, and vegetable proteins depressed (*p* < 0.001) CTTAD compared to animal proteins (main effect of protein: *p* < 0.001). In summary, pigs tolerated the extruded rice-based diets well and performed equivalently to pigs fed wheat as the sole cereal, and the use of vegetable proteins decreased the *E. coli* score.

## 1. Introduction

Postweaning diarrhoea (PWD), typically caused by the activity of enterotoxigenic and (or) enteropathogenic strains of *Escherichia coli* in the small intestine, remains a problem in commercial pig production in some parts of the world. This is exacerbated by bans or restrictions on the use of antimicrobial compounds such as medicinal zinc oxide and prophylactic antibiotics [1,2]. In association with poorer performance after weaning, the postweaning growth check significantly impacts whole-of-life productivity. A plethora of nutritional strategies exists commercially to alleviate the negative impacts of weaning in the absence of these antimicrobial compounds [3,4,5]. Among the numerous dietary options available to the industry, the choice of cereal and (or) protein sources in diets can have a major impact on PWD and performance. In this regard, rice, either as the sole cereal or in combination with others, can be used successfully in piglet diets as an alternative to other cereals such as maize, sorghum and wheat [6,7,8,9,10,11,12,13] and can reduce the incidence of PWD and decrease postweaning mortality [14]. Rice is characterised by its high starch content and varietal differences in the ratio of amylose:amylopectin, low non-starch polysaccharide (NSP) and oligosaccharide contents, and lower protein content in comparison with other cereals [15,16,17].

Newly weaned pigs are typically fed a mixture of plant (vegetable) and animal protein sources to provide amino acids and energy for body growth and development. Animal protein sources are generally used more efficiently by the weaned pig than vegetable protein sources [18] due mainly to the lack of anti-nutritive factors present and associated negative physiological effects in the gastrointestinal tract (GIT) [19,20]. Past studies have typically shown differences between individual cereal sources and protein sources on postweaning performance and aspects of GIT structure and function and PWD; however, we are unaware of studies specifically comparing different types of extruded white rice with wheat in conjunction with either animal or vegetable protein sources.

The hypotheses tested in this study were: (a) rice-based diets using animal protein sources will cause less excretion of β–haemolytic *E. coli* than diets based on vegetable proteins; (b) diets based on extruded rice will cause faster growth and less faecal excretion of β–haemolytic *E. coli* than a diet based on wheat; and (c) diets based on a medium-grain rice, with a lower amylose:amylopectin ratio, will cause better performance, irrespective of the protein source, than diets based on a long-grain rice having a higher amylose:amylopectin ratio.

## 2. Materials and Methods

### 2.1. Ethics Statement

The Murdoch University Animal Ethics Committee and the Animal Ethics and Experimentation Committee of the WA Department of Agriculture approved this experiment (Number 05-02-22).

### 2.2. Animals, Procedures, and Housing

Eighty-four entire male pigs (Large White x Landrace) aged ~24 days and weighing 6.7 ± 0.13 kg (mean ± SEM) were used in this experiment. On arrival at the Medina Research Centre, the pigs were ear-tagged, weighed, and stratified into pens of four pigs, each according to treatment and live weight. Pigs were offered their respective diets (Table 1) in groups of four for the first 7 days after weaning to accustom them to their new surroundings. For the final 2 weeks of the study, pigs were housed individually for the collection of faeces from the pen floor. Pens were of wire-mesh construction with slatted metal floors and measured 1.68 m^2^ in floor area (0.42 m^2^ per pig). Each pen was equipped with a nipple water drinker and a stainless-steel feed trough. The ambient temperature was maintained between 26 and 28 °C throughout the study using two reverse-cycle air conditioning units. The room containing the pens was cleaned daily.

### 2.3. Experimental Design, Diets, Feeding, and Sample Collection

The 21-day experiment was designed as a 3 × 2 factorial arrangement of treatments with the respective factors being (a) three cereal types, i.e., a medium-grain, lower-amylose rice (cultivar Amaroo), a long-grain, higher-amylose rice (cultivar Doongara), and wheat, and (b) two protein sources, namely vegetable and animal protein sources (Table 1). Diets are subsequently referred to as: MGAP: medium-grain rice plus animal proteins; MGVP: medium-grain rice plus vegetable proteins; LGAP: long-grain rice plus animal proteins; LGVP: long-grain rice plus vegetable proteins; WAP: wheat plus animal proteins; WVP: wheat plus vegetable proteins. None of the diets contained any antimicrobial compounds.

Isonitrogenous and isoenergetic diets were formulated to contain adequate levels of energy and nutrients for pigs of this genotype and age. Extruded rice was sourced as described previously [16] and was passed through a hammer mill to reduce particle size before incorporation into meal-based diets. The diet was offered on an ad libitum basis to piglets in mash form (Table 1). Titanium dioxide (TiO_2_) was added as an inert marker for the estimation of the coefficient of total tract apparent digestibility (CTTAD). Faecal samples were collected from the wire-mesh floor of each pig at 0800, 1000, 1200, 1400, and 1600 h on days 18–21 of the experiment. Samples collected over the 3-day period were pooled, kept at −20 °C and later thawed, mixed, freeze-dried, and ground through a laboratory hammer mill (1 mm screen) prior to chemical analysis.

### 2.4. Microbiological Assessments

Faecal swabs were taken to record initial *E. coli* presence upon arrival and then again on days 2, 5, 6, and 8 after weaning. Faecal swabs were cultured, and plates were assessed for β-haemolytic colonies displaying morphology characteristic of *E. coli* following overnight incubation, according to standard procedures [21]. The presence of β-haemolytic *E. coli* was then scored from no growth (0) to heavy colonisation (5) [21]. Piglets were monitored daily for clinical signs of diarrhoea [7]. Affected pigs (as assessed by the stockperson, who was unaware of the treatment allocation of pigs) were treated for diarrhoea by intramuscular injection with Trisoprim-480 ((trimethoprim 80 mg/mL, sulfadiazine, 400 mg/mL), 1.5 mL/30 kg body weight; Troy Laboratories, Smithfield, NSW, Australia); treatment continued until the diarrhoea ceased. Records were kept of the duration of treatment required for each treated piglet.

### 2.5. Chemical Analyses

The dry matter (DM), nitrogen (N), gross energy (GE), total starch, resistant starch (RS), amylose and amylopectin contents of extruded rice were determined as described previously [11]. Crude protein (CP) content was calculated as N × 6.25. The DM, GE, and TiO_2_ content of diets and faecal samples were determined for the estimation of CTTAD [11]. The GE content of the rice, diet, and faecal samples was determined using a Ballistic Bomb Calorimeter (SANYO Gallenkamp, Loughborough, U.K.). The TiO_2_ contents of diet and faecal samples were determined using the method described previously [22].

### 2.6. Statistical Analyses

Treatment effects were analysed by two-way ANOVA for a factorial design, with the main effects being cereal type (medium-grain rice, long-grain rice and wheat) and protein type (vegetable and animal). Average daily gain (ADG) in the first week after weaning was evaluated using the pen as the unit of replication. The ADG, average daily feed intake (ADFI) and feed conversion ratio (FCR) in weeks two and three used the individual pig as the experimental unit. For the CTTAD of DM, starch, GE, and CP, the individual pig was considered the unit of replication. All effects were considered fixed effects in the model. Fisher’s-protected least significant difference test was used (at a 5% significance level) for comparison between mean values of different variables. A *p*-value between 0.05 and 0.1 was considered a trend. All statistical analyses were conducted using the statistical package StatView 5.0 for Windows (AddSoft Pty. Ltd., Woodend, VIC, Australia).

## 3. Results

### 3.1. Antibiotic Treatments and Faecal Shedding of E. coli

Both rice-based diets fed with vegetable proteins had fewer antibiotic administrations given for clinical diarrhoea than pigs fed diets WAP or WVP, but statistically, the number of treatments was similar (*p* > 0.05) across treatments. Shedding of β–haemolytic *E. coli,* ascertained via faecal swabs, showed a trend for the main effect of protein source, with pigs fed animal proteins having a higher *E. coli* score than pigs fed vegetable proteins (0.63 vs. 0.43, *p* = 0.057). There was also a tendency for interaction between cereal type and protein source (*p* = 0.069), with this difference being associated with the greater score recorded in pigs fed diets LGAP and WAP (Table 2).

### 3.2. Production Performance

There were no statistically significant differences between treatment groups for ADG in the first week after weaning (Table 3). In weeks two and three, no main effects of cereal type on any indices were recorded (*p* > 0.05). However, pigs fed animal proteins rather than vegetable proteins were heavier (*p* = 0.01) at the end of the experiment (11.8 vs. 10.4 kg) because they grew faster (317 vs. 242 g/day, *p* < 0.001). This was a consequence of a higher ADFI (580 vs. 500 g/day, *p* = 007) and improved FCR (1.87 vs. 2.31, *p* = 0.028). No interactions (*p* > 0.05) occurred for any of the production indices (Table 4).

### 3.3. Coefficient of Total Tract Apparent Digestibility

Significant main effects for cereal type (rice vs. wheat) were observed in the CTTAD for DM, energy, and CP. Nevertheless, significant interactions occurred between cereal type and protein source for DM, energy, and CP. The CTTAD for DM was higher in pigs fed diets MGAP and LGAP than in the wheat-based diet (WAP) (0.92 and 0.92 vs. 0.85, *p* < 0.001); however, DM digestibility was similar (*p* > 0.05) between all three diets when vegetable proteins were fed to pigs rather than animal proteins (0.83, 0.82 and 0.80 for diets MGVP, LGVP, and WVP, respectively). A similar interaction occurred between cereal type and protein source for the CTTAD of energy, with diets MGAP and LGAP having the highest coefficients compared to diet WAP (0.92 and 0.91 vs. 0.83, *p* < 0.001) (Table 5).

In general, the CTTAD for starch was very high in all diets (range 0.989 to 0.999) and higher (main effect, *p* < 0.001) in both rice-based diets than in the wheat-based diets. The CTTAD for starch was higher in diet WVP than in diet WAP (0.993 vs. 0.989), which resulted in a significant interaction (*p* < 0.001). The CTTAD for CP was higher in pigs fed diet WVP compared to those fed diets MGVP and LGVP (0.76 vs. 0.67 and 0.66, respectively, *p* = 0.016) (Table 5).

## 4. Discussion

The hypotheses proposed in this study were, in general, supported by the experimental findings. Feeding vegetable proteins (as a main effect) showed a tendency to reduce faecal shedding of β-haemolytic *E. coli* in the first 8 days after weaning compared to pigs fed animal protein sources, as evidenced by the greater faecal swab score recorded in pigs fed diets LGAP and WAP. This suggests that in these two cereal sources, the presence of vegetable proteins reduced intestinal colonisation of pathogenic *E. coli*. Within a few days of piglets being weaned, enterotoxigenic and (or) enteropathogenic strains of *E. coli* may proliferate within the intestinal tract and induce diarrhoea [4]. Virtually all *E. coli* strains that cause PWD produce an alpha-haemolysin and show characteristic haemolysis on blood agar. These haemolytic strains also produce virulence factors that allow them to adhere to enterocytes (e.g., F4; F18; AIDA (Adhesin Involved in Diffuse Adherence)), and they generate one or more toxins (e.g., stable toxin (ST)a; STb; labile toxin (LT); Enteroaggregative *E. coli* heat-stable enterotoxin (EAST1)) that are responsible for inducing the diarrhoea [23]. In the current study, we did not investigate the individual attributes of the recovered *E. coli* strains but relied instead on the presence of characteristically strong β-haemolysis as a marker of strains capable of causing PWD.

Feeding extruded rice failed to translate into a reduced number of therapeutic antibiotic treatments given for clinical diarrhoea as this was statistically the same for all treatments, although it was evident that pigs fed both rice-based diets with vegetable proteins received fewer antibiotic administrations than pigs fed diets WAP or WVP. The incidence of PWD was generally low in this study, which contributed to the lack of statistically significant differences in these indices between diets. These results concur in part with those reported previously, where no correlation was found between faecal shedding of β-haemolytic *E. coli* and the number of antibiotic treatments required for PWD in the first 14 days after weaning in pigs fed either rice- or wheat-based diets [24], suggesting that while the presence and activity of enterotoxigenic *E. coli* (ETEC) are of central importance in the aetiology of PWD, dietary, and (or) physiological contributions also may have an important impact on disease expression in post-weaned piglets.

The vegetable proteins used in the current study contained considerable levels of soluble non-starch polysaccharides (NSP) and oligosaccharides, which have been shown to exacerbate the shedding of *E. coli* and cause diarrhoea [4,20,21]. Data from the present study showing an ameliorative effect of feeding vegetable proteins on faecal shedding of β-haemolytic *E. coli* suggesting that the types and quantities of dietary fibre (DF) fed in the postweaning period have a significant impact on the expression of diarrhoea. In this regard, data from the current study contrast with previous work using rice-based diets where the addition of different sources of DF increased the number of antibiotic injections required for the treatment of diarrhoea [25].

In some previous studies, increased diarrhoea and faecal *E. coli* scores have been observed in pigs fed very highly digestible rice-based diets after weaning [10,11], but the use of a (mostly) insoluble NSP source such as oat hulls was found to ameliorate the condition [10,26]. Oat hulls included in an extruded rice-based diet with animal proteins decreased total biogenic amine concentrations commensurate with lower plasma urea concentrations [26], suggesting that oat hulls were able to decrease diarrhoea where a misbalance of carbohydrate to protein entering the hindgut may occur. Alternatively, adding oat hulls to a rice-based diet might not influence fermentation behaviours in the large intestine due to its highly insoluble and lignified nature, but rather, it may have modified motility and transit time of digesta that, in turn, reduced the availability of substrate for bacterial growth [10]. While DF was added to diets in the current study either in the forms of vegetable proteins or wheat rather than oat hulls, the observations are generally consistent with the current industry philosophy that the inclusion of fibre sources in postweaning diets aids in reducing the proliferation of pathogenic bacteria such as ETEC [1,3,27,28].

Weaned piglets fed extruded medium-grain rice or long-grain rice performed similarly to pigs fed wheat in the first 3 weeks after weaning, indicating that extruded rice can replace wheat as the sole cereal in piglet diets after weaning. These data are consistent with previous studies [6,7,8,9,10,11,12,13,14] and reinforce the excellent nutritional value of rice for young pigs, especially with mild cooking that can enhance diet digestibility and ileal morphology [29]. Pigs fed rice-based diets also display lighter gastrointestinal organ weights and a greater carcase weight [11]. Vegetable protein sources in the diet depressed growth rate and feed intake and caused a deterioration in FCR in weeks two and three of the study compared to animal protein sources. These data are consistent with numerous previous observations. Weanling pigs fed a mixture of animal proteins (whey-protein concentrate and fish meal) performed better in the 2 weeks after weaning than pigs fed a mixture of plant proteins, including soybean meal, fermented soy protein and rice protein concentrate [18], commensurate with higher digestibilities of energy, DM, and CP. Moreover, post-weaned pigs fed an increased content of animal protein in an extruded rice-based diet displayed improved performance [30]. Animal proteins are more digestible than vegetable proteins [13,18,25], which are richer in anti-nutritive carbohydrate fractions, and hence more nutrients became available for body growth and development.

Seemingly in contrast to some previous findings, Montagne et al. [21] reported no differences in ADG or FCR when pigs were fed diets based on cooked (autoclaved) white rice with either animal or vegetable proteins. The reason(s) for this difference is (are) hard to explain but could be attributable to the fact that these authors infected pigs experimentally with F4:ETEC that could have disturbed the intestinal milieu associated with digestion and absorption, and (or) there was an effect of cooking form on digestibility and subsequent growth rate. Extruded rice contains very low levels of RS, whereas cooked (autoclaved) white rice that is then cooled contains approximately 20 times the RS content of extruded rice [16]. The RS level of the extruded products was not considered in the derivation of the energy value of extruded rice used in the formulation of these diets; hence, there could have been a misbalance in energy contributions between the protein sources and the extruded rice that contributed to the inferior performance of the pigs fed vegetable proteins. In this regard, Montagne et al. [21] found no difference in faecal shedding of β–haemolytic *E. coli* in pigs fed a wheat and vegetable-proteins-based diet compared to pigs fed medium-grain rice-based diets with either animal or vegetable proteins. This may also reflect the difference in the form (i.e., extrusion vs. autoclaving) of rice used in the different experiments.

Be this as it may, the presence of significant interactions between cereal and protein sources for CTTAD in the current study indicates that the dietary component responded differently to both dietary factors. For example, CTTAD for energy was significantly higher in both extruded rice-based diets irrespective of whether animal or vegetable proteins were added compared to diets WAP and WVP, whereas for CP digestion, the significant difference was caused by an apparently higher digestibility in pigs fed diet WVP compared to pigs fed diets MGVP and LGVP. It is difficult to explain the higher CP digestibility in pigs fed diet WVP compared to pigs fed diets MGVP and LGVP, given the higher DM and energy digestibilities observed in the extruded rice-based diets when vegetable protein sources were added. This might be attributable to a higher formation of microbial protein, causing an overall depressed total tract digestibility. Interpretation of total tract apparent digestibility coefficients for CP is fraught regardless because the formation of protein by the microbiota provides no real indication of the ileal digestibility of CP and absorption of amino acids.

Rice-based diets fed to piglets after weaning provide a ready form of energy in the form of glucose through starch digestion. In a previous study [11], the effects of different types of cooked white rice on starch digestion, digesta and fermentation characteristics, shedding of β–haemolytic *E. coli,* and performance after weaning were examined. Pigs received one of three rice-based diets: (i) medium-grain, (ii) long-grain, and (iii) waxy, all with animal protein sources, and a fourth diet contained mainly wheat, barley, and Australian sweet lupins. The apparent digestibility of starch measured in the ileum 14 d after weaning was highest in the medium-grain and waxy rices containing the lower amylose contents and lowest, but the same, with the other two cereal sources, similar to findings in the current study, albeit that digestibility was measured in faeces. Starch digestibility in faeces was highest in all rice diets, and digesta viscosity was highest in pigs fed the wheat-based diet in both the ileum and caecum [11].

## 5. Conclusions

Weaned piglets fed the extruded rice-based diets with either animal protein sources or vegetable protein sources performed equivalently to pigs fed either of the wheat-based diets in the first 3 weeks after weaning, confirming that extruded rice is an excellent cereal for young pigs. The use of vegetable proteins compared to animal proteins decreased production performance in weeks two and three after weaning. Vegetable protein sources displayed a trend to reduce faecal *E. coli* shedding, as evidenced by lowered faecal scores, implicating a role for implicated a role for DF sources in PWD. The CTTAD of dietary components differed according to interactions between cereal and protein sources, although the CTTAD of DM, starch and energy was generally improved by the use of extruded rice compared to wheat and was increased by the use of animal rather than vegetable proteins.

## Figures and Tables

**Table 1 animals-13-00863-t001:** Composition of the experimental diets (g/kg as-fed).

	Medium-Grain Rice		Long-Grain Rice		Wheat	
Ingredient	Animal Protein	Vegetable Protein	Animal Protein	Vegetable Protein	Animal Protein	Vegetable Protein
Rice	705.6	528.4	705.6	528.4	-	-
Wheat	-	-	-	-	780	532.8
Meat and bone meal	51.6	-	51.6	-	50	-
Whey	100	-	100	-	50	-
Bloodmeal	30	-	30	-	25	-
Fishmeal	100.4	-	100.4	-	50	-
Sweet lupins	-	100	-	100	-	100
Canola meal	-	150	-	150	-	150
Full-fat soybean meal	-	185.2	-	185.2	-	151.6
Canola oil	5	-	5	-	28.3	30
L-lysine	2.78	6.4	2.78	6.4	6.04	6.84
DL-methionine	0.36	1.12	0.36	1.12	1.2	1.47
L-threonine	1.43	2.68	1.43	2.68	2.7	2.54
L-tryptophan	0.28	0.34	0.28	0.34	0.42	0.23
Choline chloride	0.4	0.4	0.4	0.4	0.4	0.4
Dicalcium phosphate	-	18.7	-	18.7	0.7	17
Limestone	-	4.4	-	4.4	-	5.2
Salt	1	1	1	1	1	1
Premix ^1^	0.7	0.7	0.7	0.7	0.7	0.7
TiO_2_ ^2^	1	1	1	1	1	1
Calculated analysis:						
DE (MJ/kg)	15.3	15.4	15.3	15.4	15.0	15.3
CP, g/kg	200	200	200	200	197	215
SID lysine, %	1.30	1.31	1.30	1.31	1.28	1.30
Calcium %	1.2	0.8	1.2	0.8	0.91	0.8
Available P, %	0.6	0.45	0.6	0.45	0.49	0.45

^1^ Provided the following nutrients (per kg of air-dry diet): vitamins: A 1500 IU, D_3_ 300 IU, E 37.5 mg, K 2.5 mg, B_1_. 1.5 mg, B_2_ 6.25 mg, B_6_ 3 mg, B_12_ 37.5 mg, calcium pantothenate 25 mg, folic acid 0.5 mg, niacin 30 mg, biotin 75 µg; minerals: Co 0.5 mg (as cobalt sulfate), Cu 25 mg (as copper sulfate), iodine 1.25 mg (as potassium iodine), iron 150 mg (as ferrous sulfate), Mn 100 mg (as manganous oxide), Se 0.5 mg (as sodium selenite), Zn 0.25 mg (as zinc oxide) (Hogro Bronze Weaner and Grower, Rhone-Poulenc Animal Nutrition Pty Ltd., Queensland, Australia).^2^ Titanium dioxide (TiO_2_; Sigma Chemical Company, St. Louis, MO, USA).

**Table 2 animals-13-00863-t002:** Interaction means for the number of antibiotic treatments and the faecal swab score for piglets offered different diets after weaning.

Dietary Treatment		Number of Antibiotic Treatments	Faecal Swab Score ^2^
Cereal Type	Protein Source		
Medium-grain rice	Animal	1.4	0.5
	Vegetable	0.7	0.7
Long-grain rice	Animal	1.0	0.6
	Vegetable	0.6	0.3
Wheat	Animal	1	0.8
	Vegetable	1	0.3
	Pooled mean	0.9	0.5
	SEM ^1^	0.072	0.30
	Probability, *p* =		
	Cereal type	0.792	0.461
	Protein source	0.230	0.057
	Cereal × Protein	0.638	0.069

^1^ SEM: standard error of the mean. ^2^ Faecal swab score is the mean score per pig determined from swabs taken on days 2, 5, 6, and 8 after weaning.

**Table 3 animals-13-00863-t003:** Interaction means for average daily gain (ADG) of pigs kept in groups in the first week after weaning.

Dietary Treatment		Start Liveweight, kg	Liveweight after 7 Days, kg	ADG, g
Cereal Type	Protein Source			
Medium-grain rice	Animal	6.71	7.32	87
	Vegetable	6.66	7.04	54
Long-grain rice	Animal	6.68	7.31	91
	Vegetable	6.69	6.98	42
Wheat	Animal	6.97	7.46	55
	Vegetable	6.81	7.03	32
	Pooled mean	6.73	7.15	60
	SEM ^1^	0.131	0.177	110.2
	Probability, *p* =			
	Cereal type	0.900	0.991	0.625
	Protein source	0.952	0.465	0.152
	Cereal × Protein	0.996	0.984	0.903

^1^ SEM: standard error of the mean.

**Table 4 animals-13-00863-t004:** Interaction means for the performance of pigs fed different diets in weeks two and three of the experiment.

Dietary Treatment		Liveweight at Start of Week 2, kg	Liveweight at End of Week 3, kg	ADG, g	ADFI, g Day^−1^	FCR (g Feed:g Gain)
Cereal Type	Protein Source					
Medium-grain rice	Animal	7.31	11.66	311	586	1.92
	Vegetable	6.98	10.39	243	527	2.69
Long-grain rice	Animal	7.32	11.69	312	586	1.98
	Vegetable	7.04	9.97	219	488	2.32
Wheat	Animal	7.46	12.06	329	569	1.71
	Vegetable	7.03	10.71	264	485	1.92
	Pooled mean	7.15	11.07	278	540	2.13
	SEM ^1^	0.170	0.284	10.4	15.1	0.122
	Probability, *p =*					
	Cereal type	0.981	0.723	0.330	0.712	0.191
	Protein source	0.322	0.010	<0.001	0.007	0.028
	Cereal × Protein	0.987	0.942	0.803	0.874	0.545

^1^ SEM: standard error of the mean.

**Table 5 animals-13-00863-t005:** The CTTAD of dry matter (DM), starch, energy and crude protein (CP) in pigs fed different diets after weaning.

Dietary Treatment		CTTAD of:			
Cereal Type	Protein Source	DM	Starch	Energy	CP
Medium-grain rice	Animal	0.92	0.999	0.92	0.79
	Vegetable	0.83	0.998	0.82	0.67
Long-grain rice	Animal	0.92	0.999	0.91	0.78
	Vegetable	0.82	0.997	0.81	0.66
Wheat	Animal	0.85	0.993	0.83	0.78
	Vegetable	0.80	0.989	0.80	0.76
	Pooled mean	0.86	0.996	0.85	0.74
	SEM ^1^	0.006	0.0012	0.006	0.009
	Probability, *p =*				
	Cereal type	<0.001	<0.001	<0.001	0.022
	Protein source	<0.001	0.837	<0.001	<0.001
	Cereal × protein	<0.001	0.016	<0.001	0.016

^1^ SEM: standard error of the mean.

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
