# Peer review of "Differential Impacts of Cereal and Protein Sources Fed to Pigs after Weaning on Diarrhoea and Faecal Shedding of *Escherichia coli*, Production, and Total Tract Apparent Digestibility"

_animals, 2023, doi:10.3390/ani13050863_

Round 1

Reviewer 1 Report

Manuscript ID: 2189023

The manuscript is focused on a relevant topic related to an in vivo study for the evaluation of dietary strategies (based on alternative protein sources in combination with cereals) for improving gut health of weaned piglets. 

In pig livestock, PWD is a multifactorial disease of the weaning period that is responsible for economic losses and treatments with antibiotics. In the PWD pathogenesis, certainly, the nutritional approach plays a pivotal role.

Firstly, considering the complexity of E. coli pathotypes (not only ETEC and not only F4) the authors should better explain why they consider only b-hemolytic strains. The pathogenicity of E. coli is the result of the combination of many virulence factors (toxins, fimbriae, adhesive proteins, receptors…). This aspect should be considered, also in the discussion. 

Moreover, many aspects of the manuscript should be better explained as detailed below.

the experimental design and adopted methods should be better detailed in order to be properly understood and reproduced; moreover, the results exposed need to be enriched in contents, and the outcomes properly expressed in a scientific way; therefore, you should relate to more recent bibliography. 

Abstract

Please revise the length according to the journal guidelines and consider the comments on the main manuscript.

Introduction

Generally, the aim of your research should be well explained after the hypotheses (they are not the same).

Lines 53: please, consider that F4:ETEC is not the unique cause of post weaning diarrhea.

Line 58-60: please, specify this statement. Eventually add references. 

Material and methods

Please add the number of the ethical authorization.

Some details related to the experimental design are lacking: the number of animals per group, per pen. I can suppose it by reading the entire documents but it is not clear. Maybe a graphical description can help the reader (?!). Did the boxes have the same dimension? Some pens contain 3 piglets and some 4. Are they distributed in homogeneous way intra-group? 

Line 93: What about the mean weight of experimental group. Please add this information.

Line 95: Animals were housed in group for a period and then they were housed individually. Why did you adopt this protocol?

Line 110-111: which guidelines did you adopted for nutritional requirements? NRC? Please detail it. Are the diets isoenergetic and isoproteic?

Line 116: please, correct the hours, 8:00-10:00-12:00-16:00, and could you please justify the choice of this timing protocol. Why did you collect the faecal samples from the floor (not individually)?

Line 130: please, can you specify if the day 8 was the same day of the individually placement of the piglets?

Line 130: Which kind of culture medium? Please add a brief description of microbiological evaluation.

Line 133-134: Did you use a faecal score? If you did, please specify it.

Line 134-138: considering the global concern about the antimicrobial resistance, did you perform antibiogram? Did you treat the animal without a specific diagnosis? 

Line 141: please explain the acronym RS.

Line 143: please, provide references for the crude protein method.

Line 143: please, explain the acronym DM.

Line 152: please, explain the acronym FCR

Results

3.1: the shedding data are lacking. Please add this information. 

Lines 165-166: E. coli should be written in italic.

Lines 167-169: specify the unit of measurement of the values.

Lines 174-177: are the reported data related to control (?!) treatment (?!). please specify them.

3.3: did you evaluated the digestibility in all the animals included in the trial?

Line 183-188-192-193: please, specify the unit of measurement of the values and their relation with experimental groups. 

Discussion

In general, the discussion should be revised considering the obtained results and not only the state of the art. 

Lines 212-213: revise this sentence according to your aims and achievements.

Lines 218-222: Is there a statistical difference?  

Lines 225-227: please consider that hemolytic activity is not present in ETEC E. coli solely. Moreover, your results should be discussed considering the complexity of gut environment (microbiota, intestinal barrier). I know that you focalized the attention only on hemolytic bacteria and for this reason you should admit the limits of the study.

Lines 228-230: update the references for this statement.

Lines 235: what do you mean with “faecal E. coli scores”?

Lines 238-247: why did you discuss the effects of oat hulls and fiber in general in the swine diet, even if your aim didn’t include this topic?

Lines 253-254: please add references 

Lines 279-295: a more appropriate literature should be considered.

Conclusions

Revise the conclusion considering the previous comments.

Tables: 

The same units of measurements should be adopted in the tables. All the numeric results should be reported with standard deviation or standard error.

Table 1: please, could you provide chemical analyses of the feed? Moreover, specify if the percentage of the different components are expressed on the dry matter or as-fed.  Could you explain (in the manuscript) the choice of a diet with a 20% of protein? It is high. Have you considered its role in the diarrhea occurrence?

Reviewer 2 Report

The subject of the work is very current and raises important issues from the point of view of pig production. The work properly presents the research results obtained in the experiment. Unfortunately, there is one downside, rice is not found all over the world and in some climatic zones it is not possible to grow it, and the cost of delivery is too expensive. But the work is very interesting, and at the same time has the possibility of application in practice.

Introduction - the introduction very well introduces the topic of the work that will be discussed later.

Material and methods:

- why only male specimens were used in the experiment?

- as I understand it correctly, it is not explicitly written that the pigs stayed in the experiment for 3 weeks? Am I reasoning correctly?

- it is worth adding information about the difference between medium-grain and long-grain rice

Results

L176-178 - improvement in production results, including faster growth, also resulted from the fact that animal proteins are better absorbed by animals, including young animals. I think it's worth adding it here if we're starting a discussion here.

Discussion

Discussion properly conducted compared to others' other work.

Conclusions

Conclusions from the work correctly formulated and well summarizing the research carried out.

Reviewer 3 Report

The objective of the study was to estimate the effect of three types of cereal and two types of protein in the diet on growth performance, half of corn by extruded corn, raw rice or extruded rice on pig performance, diarrhoea and total tract digestibility of nutrients in young pigs. It is interesting and valuable paper. However, that manuscript needs some changes and explanation before acceptation for publication. Below you can find my suggestion for consideration:

1.      Please modify the title to fit the content of experriment. I suggest to add growth performance and use „total tract aparent digestibility of nutrients” instead of „total tract aparent digestibility”.

2.      Please use total tract apparent digestibility of nutrients instead of total tract apparent digestibility

3.      Please, avoid „strong trend”

4.      Disciption of determination of content of SID lysine, calcium, available P are missing. Results of these analysis are presented in Table 1.

5.      Please, explain all abbreviation, eg. Page 4 line 141

6.      Please, correct p-value, should be 0.01 (page 5, line 175)

7.   Please, mark the differences between cereal and protein types in all tables

'Material and methods' section should be checked and corrected. Please, add missing information on the design of the experiment.
Authors explained that dietary fibre is responsible for lower growth performance and digestibility of nutrients in pigs fed vegetable protein. Why amounts and composition of dietary fibre in experimental diets have not been analysed? Conclusion is correctly formulated.
Please, add to table values for type of protein and carbohydrates with marked statistical differences.
